# Navigational Safety Assessment of Ten-Thousand-Ton Vessels in Ship Tunnels by Ship Simulations

**Shouyuan Zhang [1], Hongyu Cheng [2], Ziming Deng [3], Lingqin Mei [1], Leyu Ding [1], Chao Guo [1], Xin Wang [1],\* and Gensheng Zhao [1],\***

[1] National Key Laboratory of Water Disaster Prevention, Nanjing Hydraulic Research Institute, Nanjing 210098, China; zhangsy@nhri.cn (S.Z.)
[2] Power China Kunming Engineering Corporation Limited, Kunming 650051, China
[3] School of Engineering, The University of Manchester, Manchester M13 9PL, UK
\* Correspondence: xwang@nhri.cn (X.W.); gszhao@nhri.cn (G.Z.); Tel.: +86-025-8582-8237 (X.W.); +86-025-8582-8215 (G.Z.)

**Abstract:** In implementing ship channels in mountainous rivers with complex topography, navigation safety has become paramount for facilitating efficient tunnel operations. The security of large vessels in tunnels has recently been the focus of a considerable amount of research in the inevitable trend of vessel upsizing. This study analyzes the characteristics of unpowered coasting distance for ten-thousand-ton bulk carriers through ship simulations. The study discovered a positive correlation between coasting length and variables including tunnel width, water depth, and sectional coefficient. Moreover, it explores the maneuvering characteristics throughout the tunnel traversal process. It proposes a vessel-following model based on car-following theory and defines different types of following distances to determine the transportation capacity of the tunnel. The research findings greatly enhance tunnel navigable safety and optimize ship tunnel operations.

**Keywords:** ship tunnel; ten-thousand-ton vessel; coasting distance; following distance; ship simulation

## 1. Introduction

Inland waterway transportation (IWT), characterized by its economic and environmental advantages, holds a pivotal position in cargo transportation, showing a swift growth tendency in recent years. In the past four decades, China's IWT has increased tenfold, increasing progressively year after year [1,2]. With the rapid economic growth and continuous expansion of global trade, IWT plays a crucial role in connecting different parts of the country and facilitating the streams of major cargoes. This phenomenon places higher demands on the efficiency, cargo capacity, and sustainability of IWT. Given the unique topography of western China, with abundant mountains and gorges, it requires additional navigational infrastructure to develop of high-grade waterway transportation [3,4]. Among various types of infrastructure, the ship tunnel is particularly suitable for application in mountainous areas, which is an engineering measure with advantages in improving navigation safety and reducing shipping routes significantly. The ship tunnel holds an excellent prospect for practical engineering implementation [5].

Ship tunnels have a long history in engineering applications. As early as 1874, the Norwegian government drafted the plan to construct the world's first full-scale tunnel for shipping, the Stad Ship Tunnel [6], to bypass the treacherous Norwegian West Coast waters, where ships frequently run aground. The tunnel was intended to connect Moldefjord Bay and Vanylvsfjord Bay directly. However, the project has yet to be completed due to various reasons, such as financial constraints and fire hazards [7,8]. Existing European ship tunnels, such as the Weilburg Tunnel, Marne Tunnel, and Malpas Tunnel, were all constructed in the last century to serve small vessels, such as yachts and sailboats [9,10]. The Silin Tunnel is one of the formally operational ship tunnels that can hold kiloton bulk carriers, linking

up the ship lift and enhancing the Wujiang River's navigation capacity in China [11,12]. As ship tunnels have limited water space and confined environments, accidents within the tunnel pose severe consequences for operational capacity, making navigation safety highly critical [13].

During a fleet's passing through a tunnel, maintaining proper spacing between vessels is essential to ensure safety [14], due to the increased risk of accidents, such as rear-end collisions in low-visibility driving environments [15]. The following distance between vessels is influenced by factors such as the braking capabilities, the handling skills of the driver, and the operating environment [15,16]. The following distance can be determined in different ways, such as experimentally, using hydraulic model tests and ship model tests [17]; numerically analyzing ship-to-ship interactions through a RANS solver in a three-dimensional model [18–21]; approximatively, by using empirical formulas [22–24]; or by combining extensive vessel handling data with probability distributions [25,26]. In addition, with the improvement of artificial intelligence and algorithms, methods based on video information perception and analysis have also emerged for determining following distance [27]. These research methods greatly enhance fleet safety in the following process.

With advancements in computer simulation technology, ship simulations integrate these factors into visual and manipulable platforms [28]. Ship simulation is a virtual emulation tool that incorporates the handling module, navigation instruments, and visual analog module, et al. It is widely utilized in channel engineering. A comprehensive range of hydrodynamic data is integrated to create a highly realistic environment [28–30]. Under the guidance of experienced pilots and operators, ship simulation can obtain accurate data on vessel attitudes and maneuvering capabilities [31]. By adjusting and calibrating the parameters, ship simulations can accurately simulate the performance of vessels under different conditions and reproduce a natural hydrodynamic environment, including water flow, waves, depth, and other factors [29]. As a result, ship simulators, such as Mobile Harbor [30] and Port of Long Beach [31], have been widely used in the design phase of various maritime projects to provide engineers with accurate recommendations and guidance. Within the virtual scenes of simulators, the operator can evaluate various navigation scenarios, which is highly beneficial for analyzing vessel handling characteristics [30–32]. Deng et al. [33] utilized a ship simulator to analyze the maneuvering characteristics of light-tonnage (50 to 3000 tons [34]) vessels in inland waterways. Yu et al. [17] combined simulation results to study the relationship between the width of the tunnel and sailing risks. Gan [35] proposed a mathematical model for light-tonnage vessels to reveal the following process.

Despite using the following processes and navigation characteristics of light-tonnage vessels with various methods, reports on large-tonnage (nearly ten thousand tons [34]) vessels in ship tunnels have been scarce. To mend this gap, this work uses ship simulations to investigate the coasting characteristics of ten-thousand-ton vessels (10,000 DWT) in ship tunnels. It discusses the effects of tunnel physical dimensions and water depth conditions on vessels' forward and reverse coasting distance. Furthermore, based on experimental data, this study proposes different types of following distances for the vessels and analyzes maneuvering characteristics throughout the tunnel traversal process. It also provides recommendations for ship tunnel design.

## 2. Methodology

### 2.1. Theory of Simulation

Motion equations can represent vessel movement in tunnels. During propulsion, the hull experiences forces such as wind, rudder, resistance, propulsion, flow resistance, and gravity. The vessel's motion is constrained by equilibrium equations in 6 degrees of freedom (DoFs), which include 3 translational DoFs (i.e., surge, sway, and heave) and

3 rotational DoFs (i.e., roll, pitch, and yaw). The equations can describe the motion state of the vessels in the aspect of acceleration, velocity, and heading, as shown as follows:

$$
\begin{cases}
m \cdot \ddot{u} + X + X_{prop} + X_{rudder} + X_{wind} = 0 \\
m \cdot \ddot{v} + Y + Y_{prop} + Y_{rudder} + Y_{wind} = 0 \\
m \cdot \ddot{\omega} + Z + Z_{prop} + Z_{rudder} + Z_{wind} + mg = 0 \\
I_x \cdot \ddot{\Phi} + L + L_{prop} + L_{rudder} + L_{wind} = 0 \\
I_y \cdot \ddot{\theta} + M + M_{prop} + M_{rudder} + M_{wind} = 0 \\
I_z \cdot \ddot{\psi} + N + N_{prop} + N_{rudder} + N_{wind} = 0
\end{cases}
\tag{1}
$$

where m is the weight of vessel, kg; X, Y, and Z represent the components of water body force on the vessel hull in the transverse, longitudinal, and vertical directions, respectively, kN; L, M, and N are the torques on the vessel to the transverse, longitudinal, and vertical axes, respectively, kN·m; u, v, and $\omega$ are velocities in the transverse, longitudinal, and vertical directions, respectively, m/s; $\Phi$, $\theta$, and $\psi$ are the rolling, pitching, and yawing angles, respectively, rad; subscripts prop, rudder, and wind indicate the propulsive, rudder, and wind forces, respectively; $\ddot{A}$ is the second derivative of the physical quantity A with respect to time; and g is acceleration of gravity, m/s$^2$.

### 2.2. Major Configurations

The ship simulation used in this study includes three mission bridges, a hardware platform simulating navigation conditions, a maneuvering simulation software platform, and a network-distributed processing compute server (Lenovo System ×3850 X6 CPU Xeon E7-4809 v3). The hardware platform is an NTPRO 5000 (Wärtsilä Marine Electronic Technology (Shanghai) Co. LTD., Shanghai, China) Full Mission Bridge Simulator provided by the TRANSAS company. It includes a triple-channel 120° horizontal view monobloc display screen, vessel steering module, maneuvering module, and navigation instruments. The virtual scenes are projected onto a 360° circular large screen, and a three-dimensional surround sound system is used to mimic a real navigational atmosphere (Figure 1). The software platform, which includes visualization software, visualization and ship model development software, and comprehensive assessment software for navigation conditions, is used for model creation, development, superimposing different flow situations, and evaluating navigational conditions. The server's role is to process and analyze data collected by the platform and system. Through these mission bridges, the officer can operate the steering gear and engine telegraph to simulate real-time procedures, achieving synchronous steerability in the actual vessel maneuvering. An experienced pilot familiar with local inland waterway routes, and a chief officer with 10 years of work experience guide the simulation, ensuring the control of the rudder, engine, and thruster. In standard vessel navigation, the officer follows the pilot's instructions to execute relevant operations.

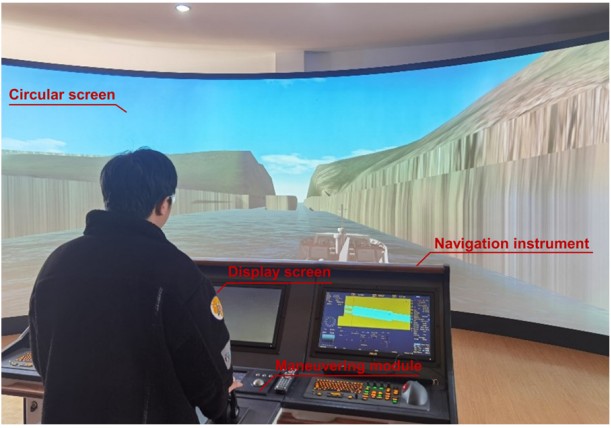

**Figure 1.** Conning the ship simulation.

### 2.3. Simulations of the Scene and the Vessel

Simulating a virtual operating scene is an important aspect of simulation settings. A realistic and accurate simulation scene allows the officer to gain practical operational experience and helps enhance the sense of control in the simulation. In this study, a designed ship tunnel in the mainstream of the Yangtze River in China was selected as the sample. The ship tunnel would span a total length of 1800 m within a mountainous terrain. It is designed to connect with the upstream reservoir via a 430 m long approach channel, whereas a 700 m anchorage area links it to downstream channels. In addition, a total of 12 hydrodynamic environments were constructed in the simulation, consisting of three tunnel dimensions and four water depths. Figure 2 illustrates the detailed three-dimensional visual scene, including the approach channel, ship tunnel, anchorage area, downstream channel, mountain body, and water body. The simulated tunnel parameters are shown in Table 1.

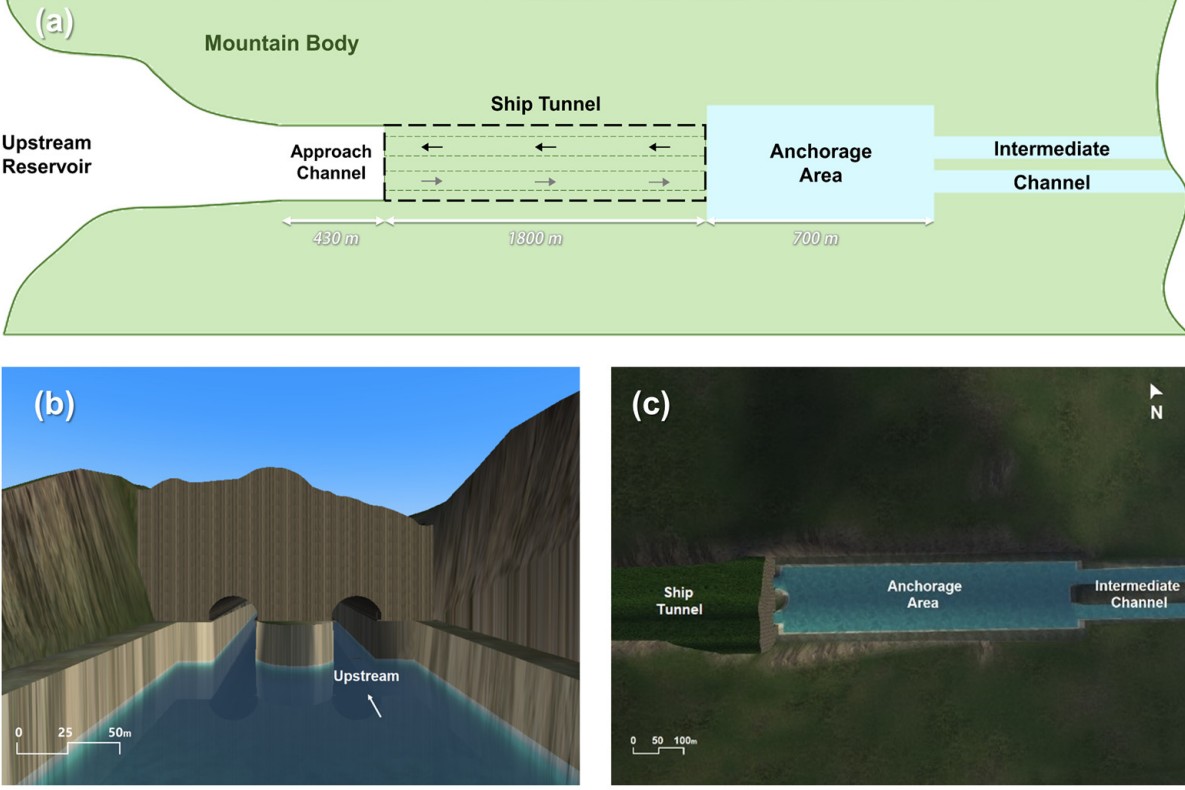

**Figure 2.** Schematic view and simulation scene of the research area. (**a**) Layout of simulated virtual environment; (**b**) three-dimensional visual scene of the ship tunnel from downstream; (**c**) aerial perspective of the ship tunnel.

**Table 1.** Specifications of the simulated tunnel.

| Parameter | Value |
| --- | --- |
| Width | 25, 28, 33.6 m |
| Length | 1800 m |
| Water depth | 8, 9, 10, 11 m |
| Lighting level | 30 lx |

A 10,000 DWT bulk carrier, which belongs to the large-tonnage vessels in IWT, was selected as the vessel sample for the experiments, and the parameters of the actual vessel were referenced in the modeling. The vessel model was specifically designed to ensure the similarity of key parameters related to vessel motion, including the coefficients of water plane and midship section, engine telegraph parameters, propeller position, propeller

parameters, number of blades, rudder characteristics, and hydrodynamic properties of the hull. The simulated vessel's specification is provided in Table 2.

**Table 2.** Specifications of the simulated vessel.

| Parameter | Value |
|---|---|
| Length overall (LOA) | 130 m |
| Breadth | 22 m |
| Draft (In ledge mode) | 5.5 m |
| Dead weight tonnage | 10,000 t |
| Type | Bulk carrier |

*2.4. Validation*

Before the formal experiment, the ship simulation must be validated and debugged to ensure that the entire system demonstrates excellent maneuverability and stability during handling, providing efficient and fitting responses to diverse hydrodynamic environments within the 6 DoFs. Moreover, the debugging process verifies that the added flow conditions align well with the responses of the vessel model, guaranteeing the accuracy and suitability of each condition in the simulation.

2.4.1. Validation of the Vessel Model

The validation test for the vessel model was conducted in a still water environment with no waves or currents at large depths. The vessel was fully loaded with a symmetrical keel. The test aims to evaluate the vessel's maneuverability and stability. Before the test, the vessel maintained a straight course at a speed of 5 kn for 3 min, with no changes to the propulsion device. At the start of the test, the steering was turned to the predetermined angle in the turn direction and then held at that angle to achieve stable circular turning. The port and starboard steering were tested once each. The test results can be seen in Table 3.

**Table 3.** Turning circle test results of the simulated vessel. ($V_C$: turning speed; $y_{090}$: transfer; $x_{090}$: advance; $D_C$: tactical diameter; LOA: length overall).

| Steering | $V_C$ (kn) | $y_{090}$/LOA | $x_{090}$/LOA | $D_C$/LOA |
|---|---|---|---|---|
| Port | 5 | 1.17 | 3.1 | 2.41 |
| Starboard | 5 | 1.13 | 3 | 2.32 |

The test data indicates that the advance during maneuvering did not surpass 4.5 times the LOA, and the tactical diameter did not exceed 5 times the LOA. When the vessel's heading changed by $10°$ to the port or starboard from the initial heading, the traveled distance was less than 2.5 times the LOA. In addition, the overshoot angle and the second overshoot angle in the $10°/10°$ and $20°/20°$ zigzag tests stayed within the limits. The vessel also exhibited good braking performance in the stopping ability test. Overall, the vessel model performed satisfactory maneuverability, meeting the current effective international maritime organization standards for vessel maneuverability [36].

2.4.2. Added Debugging Conditions

The pilot and the chief officer conducted the ship simulation multiple times after adding the conditions to ensure the current, the tunnel's visual effect, and the appropriate interactive responses between the boundary conditions and the vessel. During the debugging, although the vessel navigated through the tunnel, the limited space and narrowness of the tunnel led to higher water flow velocities between the vessel and the tunnel sidewalls, making the vessel experience lateral deviation and become susceptible to lateral drift. Based on the officer's feedback, adjustments and modifications were made to the existing database to ensure that navigation within the virtual ship tunnel database accurately represented current conditions (Figure 3).

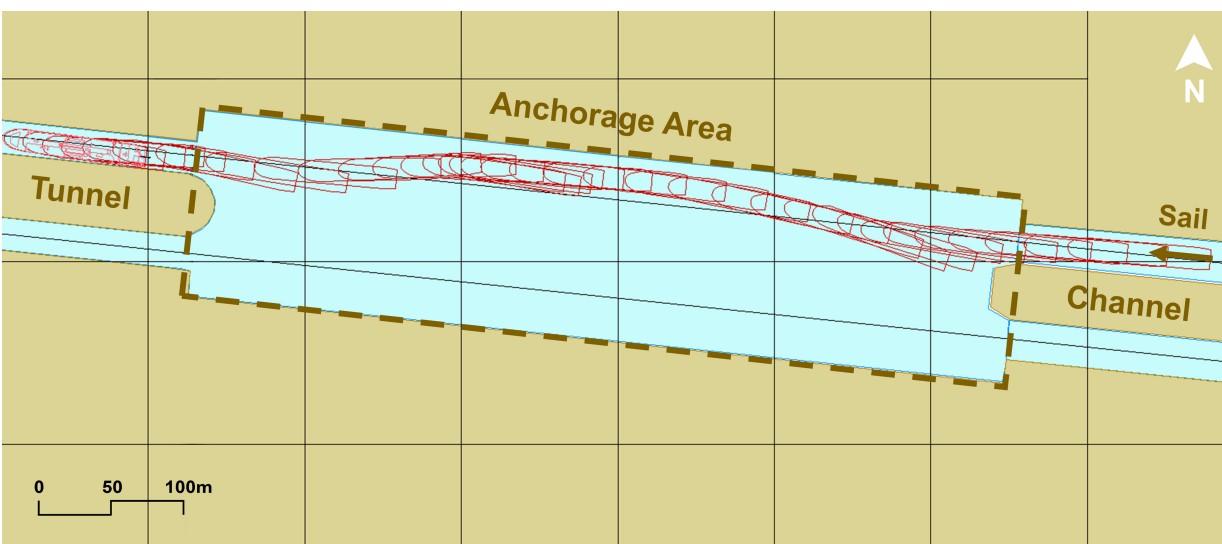

**Figure 3.** Vessel trajectory with a smooth profile following the modification of the database (The arrow indicates the direction of navigation. The vessel navigated from downstream toward the navigation tunnel, berthing in the anchorage area during the process).

After validations and modifications, the databases of various tunnel dimensions and different hydrodynamic environments are established for the formal experiments. Formal experiments include both forward and reverse coasting distance measurements. When the vessel accelerates to the expected speed, and the engines are shut down for gliding, this moment is the start point for measuring. The speed is gradually reduced due to the water resistance. The endpoint is marked when the vessel completely stops, and distance measurements are taken from the bow as the starting point.

## 3. Results

### 3.1. Forward Coasting Distance

The safe following distance of vessels in the ship tunnel is directly related to coasting distances. As the hull forces the water into the tunnel, the limited space between the sides of the vessel and the tunnel prevents the flow from returning in time, which affects the coasting performance. Therefore, the relationship between coasting distances, tunnel width, and water depth was considered. Different combined environments were simulated, including different tunnel widths (25, 28, and 33.6 m) and water depths (8, 9, 10, and 11 m). Under desirable navigation conditions in open waters, inland vessels often maintain speeds between 1.9 m/s and 2.7 m/s. However, the speed within ship tunnels decreases due to environmental driving constraints. This phenomenon was observed in China's Silin Tunnel in Wujiang River, where vessel speed is typically controlled within the range of 1.4 m/s to 1.6 m/s [33]. To maintain consistency with these conditions, an average speed over ground of approximately 1.5 m/s was maintained during the experiments. Given the tunnel length constraints, the officer could have a slight deviation in maintaining a stable speed within a short distance during the actual maneuvering, keeping the speed deviation within a 7% range.

A series of forward coasting experiments were conducted in varied tunnel widths and water depths. Each environment was repeated thrice to effectively eliminate random errors. The results are depicted in Figure 4.

Figure 4 shows the general trend that the deeper the water depth is, the longer the coasting distance will be. To clearly show the length change law of coasting distance, its relationships with water depth and tunnel width are expressed by the forward coasting distance ratio to LOA ratio in Figure 5. From a univariate perspective, when keeping the tunnel width constant, a positive correlation exists between coasting distance and water

depth. For every 1 m increase in water depth, an average increment of 16.5% in coasting distance occurs across the three tunnel widths (25, 28, and 33.6 m). This result indicates that deeper waters in the tunnel result in a more significant water level rise as the vessel traverses the tunnel, thereby reducing vessel resistance in shallow waters and enabling a greater coasting distance after the engine is stopped.

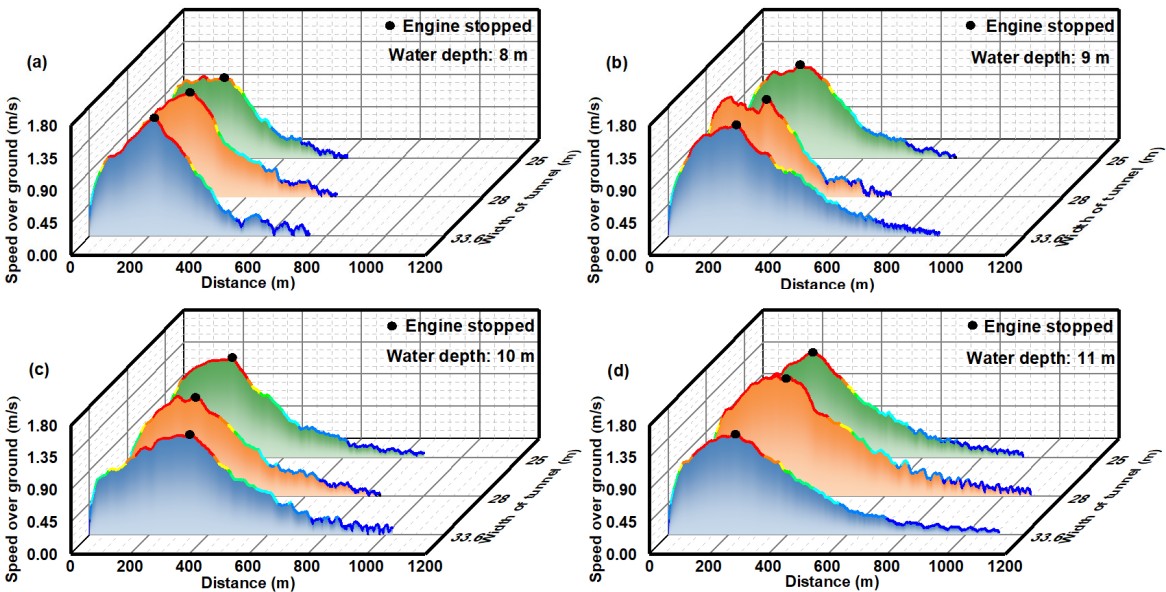

**Figure 4.** Forward coasting distance of 10,000 DWT bulk carrier after engine stopped. The water depth is (**a**) 8 m, (**b**) 9 m, (**c**) 10 m, and (**d**) 11 m. Colored lines indicate different speeds with dark red representing high speeds close to 1.5m/s and dark blue indicating low speeds close to 0.

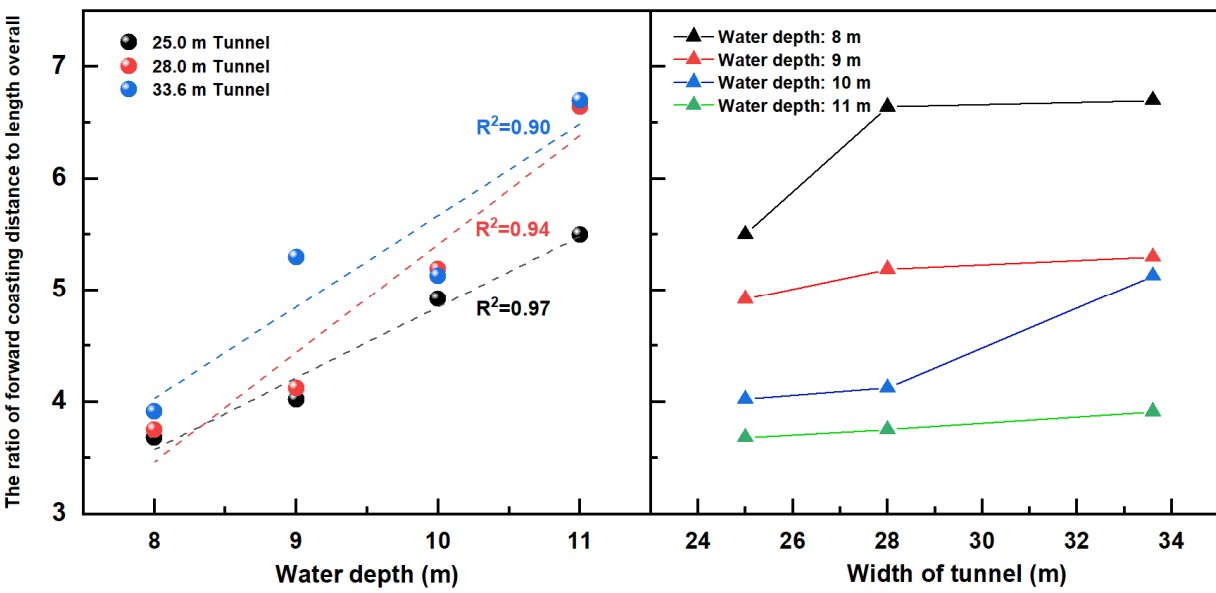

**Figure 5.** Correlation between the ratio of forward coasting distance to LOA and the water depth, and the width of tunnel.

Coasting distances increase in wider tunnels at the same water depth (Figures 4 and 5). At a water depth of 11 m, the relationship between tunnel width and coasting distance exhibits a strong linear fit, as evidenced by the high correlation coefficient of 0.97. The sudden increase in distance at a tunnel width of 33.6 m and a water depth of 10 m could be attributed to the velocity nearing 1.60 m/s. Velocity deviation results in higher coasting

distance data. Excluding these data, in terms of quantitative analysis, when the width increases by 0.45 times the vessel breadth, the coasting distance increases by 1 times the LOA on average. Furthermore, the vessel deceleration rate gradually reduces during the latter half of the stopping process, whereas the initial deceleration is relatively faster. Moreover, the acceleration during the early stage of stopping exhibits an increasing trend with wider tunnels. Specifically, the average acceleration values for tunnel widths of 25, 28, and 33.6 m are $-0.029$, $-0.053$, and $-0.061$ m$^2$/s, respectively.

### 3.2. Reverse Coasting Distance

The ship tunnel is designed for dual-channel one-way traffic. In case of sudden incidents, such as stranding or engine failure, the following vessels can reverse and retreat from the blocked tunnel to change their routes. The pilot and the officer also performed reverse coasting experiments in different hydrodynamic environments using a ship simulator, maintaining a speed of approximately 1.5 m/s when the engine stopped. The results of the experiments are presented in Figure 6. The relationships with water depth and tunnel width are expressed by the ratio of reverse coasting distance to LOA in Figure 7.

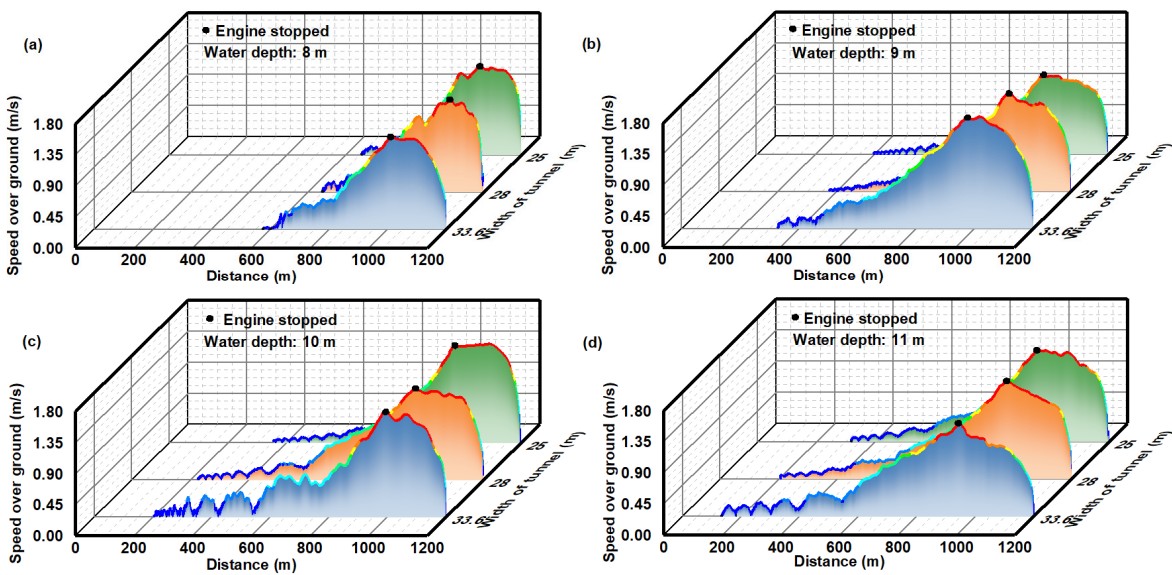

**Figure 6.** Reverse coasting distance of 10,000 DWT bulk carrier after the engine stopped. The water depth is (**a**) 8 m, (**b**) 9 m, (**c**) 10 m, and (**d**) 11 m. Colored lines indicate different speeds with dark red representing high speeds close to 1.5m/s and dark blue indicating low speeds close to 0.

Comparing the results of Figures 4 and 6 shows that the distance covered during reverse coasting is shorter compared with forward coasting. The reverse coasting distance in the 22 m wide tunnel decreases by 9.8% on average, whereas in the 33.6 m wide tunnel, it declines by more than 1.7%. This phenomenon is attributed to the fact that the block coefficient of the vessel cross-section is larger at the stern than at the bow, leading to greater flow resistance when the vessel reverses. The relationship between tunnel width, water depth, and reverse coasting distance is similar to that during forward motion, exhibiting a positive correlation (Figure 7). The linear relationship between water depth and distance is highly significant, with determination coefficients of 0.94, 0.98, and 0.99 for tunnel widths of 25, 28, and 33.6 m, respectively. When the water depth increases by 1 m, the distances in three tunnel width conditions increase by 14.7%, 19.2%, and 15.1%, respectively, with an average of 16.3%. In addition, when the tunnel width increases by 0.31 times the breadth, the distance increases by twice the LOA. Furthermore, the average values of acceleration at the initial stage of stopping for tunnel widths of 25, 28, and 33.6 m are $-0.038$, $-0.046$, and $-0.074$ m$^2$/s, respectively.

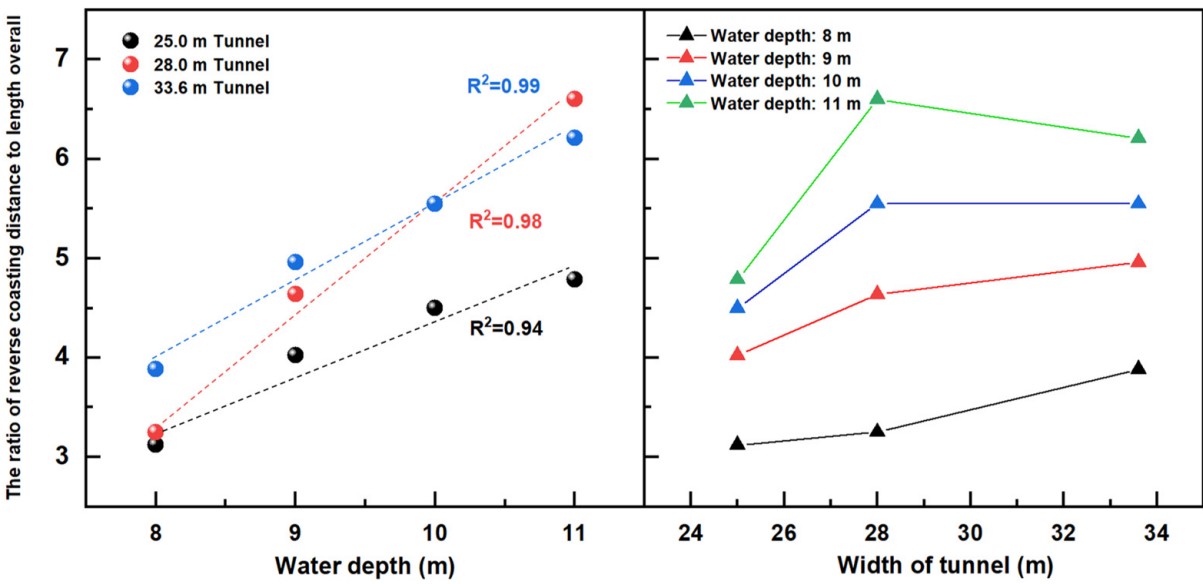

**Figure 7.** Correlation between the ratio of reverse coasting distance to LOA and the water depth, and the width of tunnel.

## 4. Discussion

### 4.1. Coasting Distance Characteristics

Many factors, including vessel dimensions, waterway conditions, and environmental considerations, influence the determination of the geometric dimensions of a ship tunnel. Wider and deeper ship tunnels improve navigation conditions. However, they also increase investment costs and construction complexities [37]. Therefore, selecting a suitable cross-sectional dimension is imperative to ensure both navigational safety and economic feasibility of construction [38]. Similarly, it is essential to note that the impact on navigational characteristics varies between wide and shallow channels and narrow and deep channels with the same cross-sectional area [37]. This necessitates the introduction of a coefficient to establish a connection between vessel dimensions, tunnel width, and water depth.

To illustrate the relationship between coasting distance and tunnel dimensions clearly, a nondimensional section coefficient n is defined as follows:

$$n = \frac{A_t}{A_v} , \tag{2}$$

where n is the nondimensional section coefficient; $A_t$ is the discharge cross-sectional area of the ship tunnel, $m^2$; and $A_v$ is the wetted cross-sectional area of the vessel, $m^2$.

Figure 8 illustrates a strong positive correlation between the forward coasting distance and the section coefficient, with a coefficient of determination of 0.70. The vast majority of data points fall within the 95% confidence band. As the sectional coefficient increases, it provides a larger space for water flow to return in the tunnel discharge cross-section. This result facilitates a more efficient dispersion of the water displaced by the bow, leading to decreased resistance on the hull and increased coasting distance. In Figure 8, bubbles of identical colors align vertically, revealing an increasing trend in each layer. This trend indicates that tunnels with narrower and deeper sectional structures for the same sectional coefficient result in longer distances of forward coasting. Furthermore, the forward coasting distance for a 10,000 DWT vessel consistently stays between 3.5 and 7.0 times the LOA within the sectional coefficient range of 1.6–3.2.

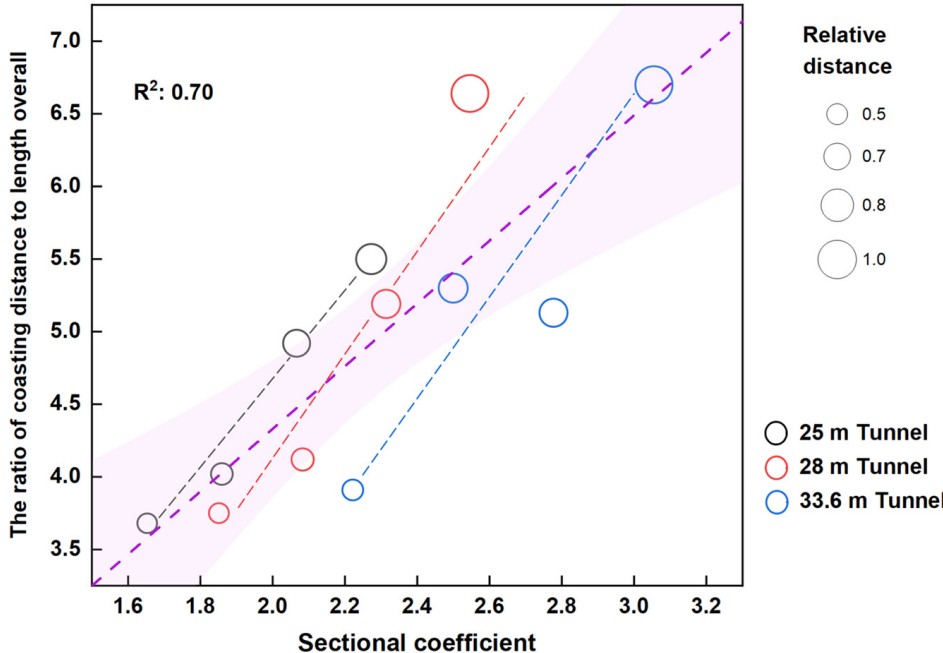

**Figure 8.** Correlation between the ratio of forward coasting distance to LOA and the sectional coefficient (The light purple area represents the 95% confidence band).

Figure 9 shows that a positive relationship exists between the ratio of reverse coasting distance to LOA and the sectional coefficient. Most of the data points fall within the 95% confidence band, exhibiting a strong linear relationship with a coefficient of determination of 0.90. For the identical sectional coefficient, the reverse coasting distance in a narrow tunnel is shorter, which is basically the same as the forward coasting trend. Moreover, within the sectional coefficient range of 1.6–3.2, the reverse coasting distance of 10,000 DWT vessels stays between 3.0 and 6.5 times the LOA.

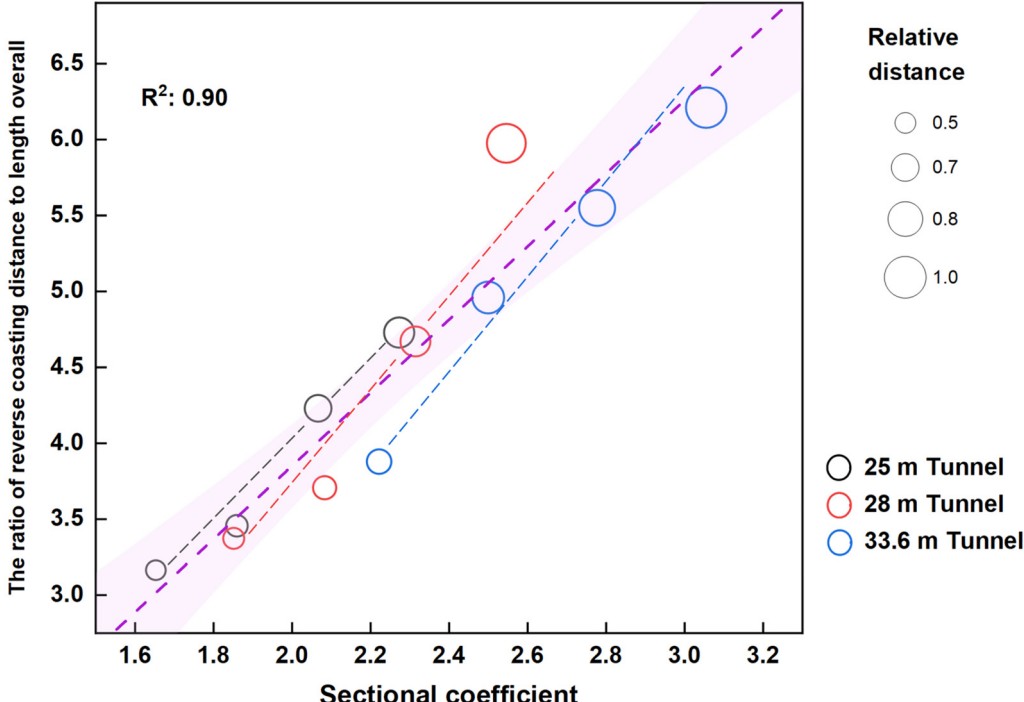

**Figure 9.** Correlation between the ratio of reverse coasting distance to LOA and the sectional coefficient (the light purple area represents the 95% confidence band).

### 4.2. Following Distance

#### 4.2.1. Vessel-Following Model

Based on the data from coasting experiments, the car-following theory is introduced to analyze the safe following distance for 10,000 DWT bulk carriers navigating in ship tunnels. A vessel-following model derived from this theory can effectively reveal complex traffic behaviors, aiding in the analysis of tunnel passing capacity [39]. This model accounts for the dynamic following process within a single tunnel, where overtaking is strictly prohibited, and the following vessel must leave space for the preceding vessel. This characteristic is similar to the process of vehicle following on a one-way street. The vessel-following model of a fleet with the same displacement is shown in Figure 10.

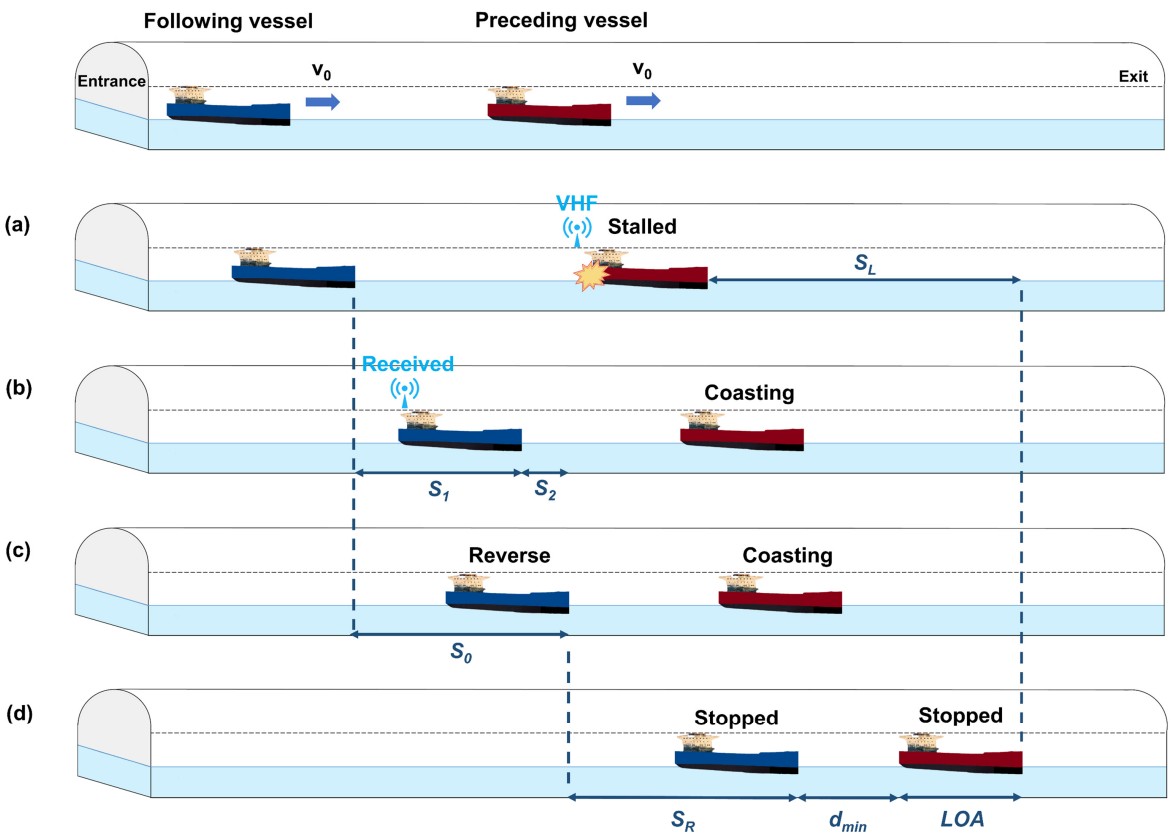

**Figure 10.** Schematic of vessel-following model. (**a**) Engine stalled in the preceding vessel; (**b**) following vessel receiving notification; (**c**) following vessel reversing; (**d**) preceding and following vessels completely stopped.

Figure 10a illustrates that, when the engine failure or stalling occurs in the preceding vessel, it enters a powerless coasting state until it stops completely, and the malfunction information is sent via a very high frequency (VHF) broadcast. The total coasting distance of the preceding vessel is denoted as $S_L$. In comparison with realizing the forward malfunction of the following vessel by the automatic identification system (AIS), which commonly has a delay of 5–30 s in updating information for inland vessels [40], broadcasting the notification through VHF radio has a faster response. The distance traveled by the following vessel at a constant speed from the start of the VHF broadcast until the operation of the reversing by the following vessel's officer is denoted as $S_0$. Figure 10b shows the distance covered by the following vessel during the broadcasting, labeled as $S_1$. Figure 10c illustrates the traveled distance during decision making and braking operation of the following vessel, marked as $S_2$. In the final stage, as shown in Figure 10d, the preceding and following vessels stop completely. The distance traveled by the following vessel during the reverse braking is denoted as $S_R$, and the minimum safe distance maintained between the vessels

after coming to a stop is $d_{min}$. Therefore, the following distance $S_F$ in the ship tunnel can be expressed as follows:

$$S_F = S_0 + S_R + d_{min} + LOA - S_L, \tag{3}$$

and

$$S_0 = S_1 + S_2 = v_0 t = v_0(t_1 + t_2), \tag{4}$$

where $v_0$ is the speed over ground, m/s; t is the total response time, s; $t_1$ is the duration of VHF broadcasting, s; and $t_2$ is the duration of decision making and braking operation, s.

The above formula represents the distance maintained between the two 10,000 DWT vehicles, which is referred to as the minimum following distance (MFD) when the preceding vehicle stalled. When the preceding vessel has a larger displacement than the following vessel, the braking distance of the preceding vessel is longer, which is safer and abundant for braking. Therefore, more attention should be paid to the situation where the braking distance of the preceding vessel is significantly shorter than that of the following vessel. In Formula (3), if $S_L$ is replaced by the reverse stopping distance of the smaller displacement vessel $S_{RL}$, $S_F$ will represent the following distance with significant displacement differences (SDDFD). When the vessels with the same displacement reverse braking together, $S_L$ is equivalent to $S_R$. In this case, $S_F$ is referred to as the general following distance (GFD). When the preceding vessel suddenly stops with grounding, the total coasting distance of the preceding vessel is 0. In this case, $S_F$ is referred to as the abundant following distance (AFD). For the various scenarios mentioned above, $S_F$ can be expressed as

$$S_F = \begin{cases} S_0 + S_R + d_{min} + LOA - S_L & \text{MFD} \\ S_0 + S_R + d_{min} + LOA - S_{RL} & \text{SDDFD} \\ S_0 + d_{min} + LOA & \text{GFD} \\ S_0 + S_R + d_{min} + LOA & \text{AFD} \end{cases} \tag{5}$$

If an accident occurs on the preceding vessel, such as grounding, collision with tunnel walls, or capsizing, and the site cannot be cleaned quickly, then the following vessel should reverse and exit the tunnel from the entrance. In this study sample, the following vessel can reverse to the anchorage area and then turn into another tunnel to go through. Similar to forward coasting, when the preceding vessel reverses into the anchorage area, it can choose to coast or reverse to stop. The reverse following distance $S_{BF}$ can be also expressed by the vessel-following model. $S_{BF}$ for MFD, SDDFD, and GFD can be expressed as follows:

$$S_{BF} = \begin{cases} S_0 + S_{RB} + d_{min} + LOA - S_{LB} & \text{MFD} \\ S_0 + S_{RB} + d_{min} + LOA - S_{RLB} & \text{SDDFD}, \\ S_0 + d_{min} + LOA & \text{GFD} \end{cases} \tag{6}$$

where $S_{LB}$ is the reverse coasting distance of the preceding vessel, m; $S_{RB}$ is the reverse stopping distance of the following vessel, m; and $S_{RLB}$ is the reverse stopping distance of the preceding vessel, m.

4.2.2. Quantitative Analysis

The following distance is quantitatively analyzed based on the vessel-following model combined with the coasting data of the ship simulation. Time delay is an important parameter in the following theory [39,41]. Similarly, the duration of VHF broadcasting, decision making, and braking operation in vessel following must be determined. According to Ming [42], the total response time to estimate the accident situation ahead by AIS is about 90 s; the time via VHF broadcast is shorter than this value. Gan [35] noted that the broadcasting duration of inland vessels is about 10–30 s. Given that safety is crucial in tunnel navigation, the value of $t_1$ is taken as 30 s. On the basis of the actual operation of ship simulation, decision making and braking operation take the pilot and the officer about 30 s. Consequently, the total response time is determined as 60 s. Moreover, given the

sufficient space between the vessels for maneuvering, the minimum safe distance is set as the LOA [42]. In the design of inland ship tunnels for 10,000 DWT vessels, research suggests that maintaining a tunnel discharge cross-sectional area equal to about 1.86 times the wetted cross-sectional area of the vessel is considered safe and economical [43]. In addition, to reduce blockage effects, the water depth should be greater than 1.5–1.6 times the draft to avoid inhibiting the backflow at the bottom of the tunnel [42–44]. According to the above requirements, the following distance is calculated with the speed of 1.5 m/s based on the four most appropriate tunnel dimensions. The calculation of SDDFD simulates the scenario where a 10,000 DWT bulk carrier follows a thousand-ton (1000 DWT) bulk carrier. The navigation parameters of the 1000 DWT vessel are referenced from the literature [35,37], with $S_{RL}$ and $S_{RLB}$ values of 30 m and 26 m, respectively. The calculated results for the following distances are shown in Table 4.

**Table 4.** Following distances in the ship tunnel.

| Heading | Tunnel Width (m) | Water Depth (m) | Sectional Coefficient | $S_R$ (m) | $S_L$ (m) | MFD /LOA | GFD /LOA | SDDFD/LOA | AFD /LOA |
|---|---|---|---|---|---|---|---|---|---|
| Forward | 25 | 9 | 1.86 | 420 | 522 | 1.90 | 2.69 | 5.69 | 5.92 |
| | 28 | 9 | 2.08 | 425 | 536 | 1.84 | 2.69 | 5.73 | 5.96 |
| | 28 | 8 | 1.85 | 390 | 487 | 1.94 | 2.69 | 5.46 | 5.69 |
| | 33.6 | 8 | 2.22 | 403 | 508 | 1.88 | 2.69 | 5.56 | 5.79 |

| Heading | Tunnel Width (m) | Water Depth (m) | Sectional Coefficient | $S_{RB}$ (m) | $S_{LB}$ (m) | MFD /LOA | GFD /LOA | SDDFD /LOA | |
|---|---|---|---|---|---|---|---|---|---|
| Astern | 25 | 9 | 1.86 | 342 | 522 | 1.30 | 2.69 | 5.12 | |
| | 28 | 9 | 2.08 | 399 | 599 | 1.15 | 2.69 | 5.56 | |
| | 28 | 8 | 1.85 | 264 | 422 | 1.47 | 2.69 | 4.52 | |
| | 33.6 | 8 | 2.22 | 330 | 504 | 1.35 | 2.69 | 5.03 | |

The results show that as the tunnel width increases, the MFD values for both forward and reverse ship motions decrease, while the SDFD and AFD increase. The SDFD and AFD increase proportionally with water depth, whereas the MFD exhibits a negative correlation with depth. The relationship between MFD and tunnel dimensions was also confirmed in Deng's study on 1000 DWT vessels [35]. Furthermore, GFD is independent of tunnel width and water depth according to its definition. The MFD and AFD of 10,000 DWT bulk carriers' fleets are 1.84–1.90 times the LOA and 5.79–5.96 times, respectively. In comparison with the current standard that the fleet following distance of the ship tunnel located in the Wujiang River in China is not less than 100 m (approximately 1.79 times the LOA of the design vessel type), the MFD results calculated by the vessel-following model are consistent with the reality. Moreover, the GFD is independent of braking and is calculated as 2.69 times the LOA. The SDDFD is slightly smaller than the AFD for the fleet, ranging from 5.46 to 5.69 times the LOA. Taking the maximum value of MFD as 1.90 times the LOA, the maximum passing capacity of the tunnel is 5.4 10,000 DWT bulk carriers passing through simultaneously in a single direction. If the AFD is calculated at 5.96 times the LOA, then the tunnel can guarantee 2.8 vessels passing at least. Normally, the tunnel accommodates 4.5 vessels (in GFD). In exceptional cases, when reversing through the tunnel, the MFD is 1.15–1.47 times the LOA. The SDDFD should be used as the AFD for a safety perspective during reverse passing, with a maximum value of 5.56 times the LOA.

### 4.3. Handling Characteristics

While managing ship simulation, a significant focus was placed on the maneuvering characteristics during the vessel's entry and exit from the tunnel and its behavior during long-distance tunnel navigation. The overall operation was smooth when departing from the anchorage area. However, in the range of 50 m near the tunnel entrance, a heading

deviation trend was noted, necessitating vigilant course control during this phase. Upon tunnel entry, the surrounding environment abruptly shifted from bright to dark, requiring the officer's pupils time to adjust to this new environment, leading to a physiological phenomenon of dark adaptation lag [45]. Similarly, the tunnel's external light affected driving safety upon exit, demanding attentiveness to avoid excessive glare. To mitigate driving interference caused by light changes, the luminance at the tunnel entrance or exit can be increased appropriately, and the light-blocking measures, such as a light baffle, can be set outside the tunnel to reduce the gap between internal and external brightness.

In the tunnel, the vessel should travel along the center line of the tunnel as far as possible, to allocate an equal water area on both sides of the hull and achieve a symmetric flow field, thereby avoiding course deviation due to asymmetrical forces [35]. The vessel encountered greater flow resistance when the tunnel width was 25 m, causing a minor speed decline between 450 m and 650 m from the entrance. After traversing this area, the course gradually stabilized, and the speed remained steady. When the tunnel width increased to 28 m, vessel maneuvering noticeably eased, and at 33.6 m width, abundant space was on both sides, resulting in less resistance compared with the 25 m width. Furthermore, it took about 21–23 min to cross the tunnel. Handling in low visibility conditions for a long time often leads to physiological fatigue, and the lack of reference objects inside the tunnel heightens collision risks. Therefore, distance markers should be installed along the course to prevent officers from making distance estimation errors due to decreased visual perception. In addition, to assist the vessel in navigating along the center line, a light belt is suggested to facilitate the judgment of the tunnel's mid-axle position. Within 600 m of the exit, the vessel began to experience a gradual reduction in resistance, causing a slight speed increase. Near the tunnel outlet, the flow field on both sides ceased to be symmetrical, and the heading deviated again. Therefore, appropriate deceleration measures must be implemented in advance when sailing out of the tunnel.

Safe navigation and comprehensive tunnel traffic supervision are essential prerequisites for improving the passage capacity of tunnels [46]. Human error by vessel operators is the leading factor in tunnel safety problems and a significant obstacle to tunnel operation [13]. The previous risk assessment literature has shown that overspeed is the leading cause of tunnel safety accidents [47,48]. In addition, tunnel supervision capacity and emergency response capability are the key factors to ensure efficient tunnel operation [13]. Therefore, in actual operations, real-time monitoring of vessel speed and following distance, timely warning and adjustment instructions for very close following situations, and effective tunnel accident emergency plans are essential to ensure tunnel navigation safety and improve passage capacity.

## 5. Conclusions

This paper compares and analyzes forward and reverse coasting distances of ten-thousand-ton bulk carriers in ship tunnels of different dimensions by ship simulation. Furthermore, a vessel-following model suitable for ship tunnels is proposed. The model provides definitions and categorizations of different types of following distances, including those for vessels with significant differences in displacement. Based on the ship simulation data, the one-way passing capacity of the tunnel sample is calculated and discussed. Additionally, the handling characteristics are detailed through ship maneuvering simulations. Crucial navigation points and relevant recommendations are also described. It can be concluded that:

- Ship simulation can effectively simulate vessel navigation in inland waterway ship tunnels and accurately reflect forward and reverse coasting characteristics.
- For a 10,000 DWT bulk carrier with a speed of 1.5 m/s, an increase of 1 m in water depth results in 16.5% (16.3% in reverse) average increase in forward coasting distance. An increase of 0.45 (0.31) times the vessel breadth leads to a one-time increase in the LOA for the forward (reverse) coasting distance. The forward and reverse coasting

- distances are positively correlated with the tunnel water depth or width. On average, the reverse coasting distance is 10% shorter than the forward distance.
- The acceleration of the vessel during the initial phase of stopping is positively correlated with tunnel width. Forward and reverse coasting distances have a strong linear relationship with the sectional coefficient, with correlation coefficient values of 0.7 and 0.9, respectively.
- In forward motion, the MFD and AFD of 10,000 DWT bulk carriers' fleet are 1.84–1.90 times the LOA and 5.79–5.96 times the LOA. And the GFD is 2.69 times the LOA. The SDDFD is slightly smaller than the AFD, ranging from 5.46 to 5.69 times the LOA. During reversing, the maximum MFD and SDDFD values are 1.47 and 5.56 times the LOA, respectively.
- Course deviation phenomena occur near the entrance and exit of the tunnel, and changes in lighting can affect the officer's visual perception. Upon entering the tunnel, the vessel's speed slightly reduces due to increased flow resistance, with a modest speed increase near the tunnel exit. The pilot and officer should control the speed and navigate along the centerline of the tunnel throughout the course. The implementation of distance and lighting markers indicating the tunnel center line is recommended for ship tunnels.

**Author Contributions:** Conceptualization, S.Z. and Z.D.; methodology, Z.D. and X.W.; software, C.G.; validation, H.C., C.G. and S.Z.; formal analysis, S.Z.; investigation, L.M. and Z.D.; resources, L.M. and L.D.; data curation, S.Z. and H.C.; writing—original draft preparation, S.Z.; writing—review and editing, S.Z.; visualization, S.Z. and L.D.; supervision, G.Z. and X.W.; project administration, G.Z. and X.W.; funding acquisition, G.Z. and X.W. All authors have read and agreed to the published version of the manuscript.

**Funding:** This research received no external funding.

**Data Availability Statement:** Data available on request due to restrictions, e.g., privacy or ethical. The data presented in this study are available on request from the corresponding authors.

**Acknowledgments:** The authors would like to thank Yuhong He and Officer Jialiang Lang for their assistance and technical guidance in this study.

**Conflicts of Interest:** The authors declare no conflict of interest.

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
