# Peer review of "Navigational Safety Assessment of Ten-Thousand-Ton Vessels in Ship Tunnels by Ship Simulations"

_water, doi:10.3390/w15203584_

Round 1

Reviewer 1 Report

The manuscript used a ship simulation system to investigate the navigational characteristics of a ten-thousand-ton ship in an undersea tunnel. The effects of the physical dimensions of the tunnel and the water depth on the ship's glide distance were discussed, and different types of ship-following distance models were proposed. Overall, the study has an average depth of discussion, but the workload is appropriate and innovative. I suggest that it is acceptable for publication after minor revision.

Comments:

1.      Consistent with Figure 5, adding a figure for the correlation of reverse coasting distance with water depth and tunnel width is suggested.

2.      Could the author please briefly clarify in the text the role of the section coefficient and why it should be evaluated?

3.      Check whether the "Figure 6" described in the text in the second paragraph of Section 4.2 is wrong. According to the meaning of the text, it should be "Figure 7".

4.      In Section 4.1, the authors show that as the tunnel width decreases, the forward coasting distance tends to increase for a given section coefficient. This contradicts the conclusions in section 3.1.

5.      In section 4.2, the authors calculated the following distance of the vessel under different working conditions. It is suggested that authors discuss the relationship between following distance and tunnel width and water depth. When conditions permit, tunnel size and water depth are introduced into the vessel-following model.

6.      As the author stated, studying the following distance between large vessels is crucial for ensuring safety and improving the efficiency of tunnel passage. Section 4.3 should discuss more about how to increase the passage capacity of tunnels while ensuring safety.

7.      There is no need to reiterate the significance of the research in the conclusion.

The English language still needs further polishing and modification, especially the preface and the conclusion.

Author Response

Dear Sir or Madam, thank you for your valuable advice on our manuscript. We have read your comments carefully and performed relevant revisions. The manuscript highlighted the revised contents, and manuscripts with/without revision tracks were uploaded separately in the attached files. The detailed point-by-point response to your great comments was also uploaded in the attached file. Due to the addition and order changes of figures, we uploaded all figures in the right order.

Reviewer 2 Report

Ship and ships in tunnel are superb subjects.

The paper lacks could be the followings.

Coasting and reverse coasting distances are not defined. It could be included end section 2.

Fluid motion is not detailed. Backflow, equal area, symmetry are not introduced before sections 4.2 and 4.3. Water velocity underground the hull, added water current by ship may be detailed.

Tunnel trafic supervision is not mentioned.

The paper presents numerous results and valuable discussion without sufficient definitions.

References are usefull. Edit names refs 8 41 42.

« Unrestricted » second line section 241 is unclear.

Emmission, gates, fauna, safety quay are parts of ship tunnel incertitudes.

Author Response

Dear Sir or Madam, thank you for your valuable advice on our manuscript. We have read your comments carefully and performed relevant revisions. The manuscript highlighted the revised contents, and manuscripts with/without revision tracks were uploaded separately in the attached files. The detailed point-by-point response to your excellent comments was also uploaded in the attached file. Due to the addition and order changes of figures, we uploaded all figures in the correct order.

Round 2

Reviewer 1 Report

The manuscript used a ship simulation system to investigate the navigational characteristics of a ten-thousand-ton ship in an undersea tunnel. The effects of the physical dimensions of the tunnel and the water depth on the ship's glide distance were discussed, and different types of ship-following distance models were proposed.

Overall, the quality of the manuscript has significantly improved after the first round of revisions, and the modifications are more than satisfactory. However, I still have some suggestions regarding the quality of the figures in the manuscript.

Comments:

1.      The figures need to be further embellished and improved in clarity.

2.      It is suggested to label the parts of the ship simulation platform in Figure 1.

3.      The superscripts (a), (b), and (c) in Figure 2 should be uniformly colored.

4.      It is recommended that the data for vertical comparisons in Figure 8 be labeled.

Minor editing of English language required!

Author Response

Dear Sir or Madam, thank you for your valuable advice on our manuscript. We have read your comments carefully and performed relevant revisions. The manuscripts with revision tracks were uploaded separately in the attached files. The detailed point-by-point response to your comments was also uploaded in the attached file.

Reviewer 2 Report

The revision of the paper gives valuable details and explanations. These two items could be added for clearness:

-       - Draft 5.5 m in Table 2 may corresponds to vessel in ledge mode instead on load (near 8 m). It could be indicated.

-       - “extensive” deep-water conditions could be clarified. Is-it means many different water depths or small and large depths ?

The paper could be published in the present form.

Author Response

Dear Sir or Madam, thank you for your valuable advice on our manuscript. We have read your comments carefully and performed relevant revisions. The revised contents were highlighted in yellow, and manuscripts with revision tracks were uploaded separately in the attached files. The detailed point-by-point response to your comments was also uploaded in the attached file.
